# Effect of Pd/ZnO Morphology on Surface Acoustic Wave Sensor Response

**DOI:** 10.3390/nano11102598

**Published:** 2021-10-02

**Authors:** Dana Miu, Izabela Constantinoiu, Cornelia Enache, Cristian Viespe

**Affiliations:** 1Laser Department, National Institute for Laser, Plasma and Radiation Physics, Atomistilor 409, RO-077125 Magurele, Romania; dana.miu@inflpr.ro (D.M.); izabela.constantinoiu@inflpr.ro (I.C.); cornelia.sima@inflpr.ro (C.E.); 2Faculty of Applied Chemistry and Materials Science, University Politehnica of Bucharest, RO-011061 Bucharest, Romania

**Keywords:** ZnO, Pd, morphology, pulsed laser deposition, hydrogen, surface acoustic wave, sensor, bilayer, nanoporous

## Abstract

Laser deposition was used to obtain Pd/ZnO bilayers, which were used as sensing layers in surface acoustic wave (SAW) sensors. The effect of laser deposition parameters such as deposition pressure, laser energy per pulse, laser wavelength or pulse duration on the porosity of the Pd and ZnO films used in the sensors was studied. The effect of the morphology of the Pd and ZnO components on the sensor response to hydrogen was assessed. Deposition conditions producing more porous films lead to a larger sensor response. The morphology of the ZnO component of the bilayer is decisive and has an influence on the sensor properties in the same order of magnitude as the use of a bilayer instead of a single Pd or ZnO layer. The effect of the Pd film morphology is considerably smaller than that of ZnO, probably due to its smaller thickness. This has implications in other bilayer material combinations used in such sensors and for other types of analytes.

## 1. Introduction

Due to its energy potential, hydrogen is the initiator of a new generation of fuel. Compared to fossil fuels, the environmental impact of hydrogen is low [1,2]. Hydrogen is used in fields such as aerospace programs, the automotive industry, pharmaceutics, the chemical industry, metallurgy, etc. [3,4,5]. However, the use of hydrogen also requires strict safety measures, because, at concentrations over 4%, hydrogen becomes flammable [6,7]. Thus, because it is used on a large scale, it is necessary to develop methods to prevent disasters. In order to achieve this, sensors for the detection of hydrogen at low concentrations remain a safe and accessible method.

Surface acoustic wave (SAW) sensors are notable due to their very high sensitivity, short response and recovery times, the possibility of wireless operation, small size and ease of fabrication [8,9]. The principle of operation of this type of sensor is based on the transformation of an electrical signal into a mechanical wave that travels at the surface of the sensor, which includes a sensitive area. When the gas molecules to be detected appear at the level of the sensitive area, the mechanical waves are disturbed, thus changing the oscillation frequency. The surface acoustic wave is then transformed back into an electrical signal, with an output oscillation frequency shifted from the input one. The frequency shift is mainly caused by the mass or acoustoelectric changes that take place at the level of the sensitive layer in the presence of gas molecules [10]. SAW sensors are used both for the detection of hydrogen and for the detection of volatile organic compounds, ammonia, hydrogen sulfide or other toxic or explosive gases [8,11,12,13,14]. 

One of the main factors that affects their performance is the sensitive material of the sensor. Depending on the type of gas to be detected, both the type and the morphology of the sensitive material are important. For the detection of hydrogen, SAW sensors with different types of sensitive layers have been developed: metal oxide semiconductors (ZnO, SnO_2_, TiO_2_) [15,16,17,18], metals (Pd and Pt) [19] and composite materials [9,13,20]. The morphology of these materials has a very important role in ensuring the interaction of gas molecules with the sensitive film over a large surface. Therefore, the synthesis of sensitive layers with optimal porosity is pursued. 

Pulsed laser deposition (PLD) is one of the most efficient methods for the synthesis of thin films. It allows the deposition of a wide variety of materials with the advantage of maintaining their stoichiometry [21]. It is a reproducible method, which allows deposition at low temperatures and ensures purity [22,23]. In addition, due to the possibility to vary a wide range of deposition parameters, different types of morphologies can be obtained, which is a very important aspect for the field of sensors [16]. Such important deposition parameters are: the gas pressure inside the deposition chamber, the wavelength and repetition rate of the laser, the distance between the target and the substrate, the number of pulses and pulse duration and the substrate temperature [24].

In a previous paper on SAW sensors with Pd/ZnO bilayers, we proved that the bilayer leads to better sensor properties than single Pd or ZnO layers having the same total thickness [25]. In the present research, we varied the morphology (porosity) of the Pd and ZnO layers in order to assess the effect on the response of the SAW sensors. The morphology of the Pd and ZnO layers was modified by varying various laser deposition parameters, such as pressure, laser power and wavelength, number of pulses and pulse duration. As mentioned, the morphology of the sensitive films, especially the porosity, has a large effect on the response of the SAW sensor to gases.

## 2. Materials and Methods

ZnO and Pd layers and Pd/ZnO bilayers were deposited onto Si and ST-X quartz substrates using PLD. A Nd-YVO_4_ laser (Lumera Rapid, Kaiserslautern, Germany) with 10 ps pulse duration was used for the depositions, at various power levels (0.1; 0.2; 0.3 W). The laser beam was focused using a 320 mm lens onto the surface of the targets placed in a vacuum chamber equipped with a gas pressure controller (MKS 600, Munchen, Germany) and a mass-flow control system (MKS multigas 647, Munchen, Germany). The ZnO films were deposited in oxygen in order to ensure correct stoichiometry. The Pd films were deposited in argon. Both in the case of ZnO and of Pd, the deposition pressures used were between 100 and 700 mTorr. The ablated target material was deposited onto substrates placed 4 cm from the targets, parallel to the target surface. In order to avoid target erosion, which leads to unfavorable film morphology, the targets were subjected to continuous movement during deposition by means of computer-controlled x-y tables. More details of the deposition system can be found in [17]. Two laser emission wavelengths were used: 532 nm and 1.06 µm. Most of the films were deposited using a 10 kHz laser repetition rate. In order to ascertain the possible effect of the laser repetition rate on the deposition, some depositions were made at a 50 kHz repetition rate. Some depositions were also made using a Nd-YAG laser with ns pulse durations (EKSPLA NL301HT, Ekspla, Vilnius, Lithuania) to determine the effect of pulse durations on the deposited film morphology. 

Thin film monolayers and bilayers were deposited onto Si substrates for scanning electron microscopy (SEM) analysis. In the case of the bilayers, the Pd film component was deposited with a smaller number of pulses, being thinner than the ZnO component. The Pd layer must be thinner in the sensor, because its role is to dissociate the H_2_ molecule, so that the resulting H atoms have larger mobility, increasing their detection after diffusion into the ZnO component [26]. FEI QUANTA SEM (Hillsboro, OR, USA) for surface analysis and Thermo Scientific Apreo SEM (Thermo Scientific, Waltham, MA, USA) for cross-section analysis were used to study the dependence of the film morphology on deposition parameters including deposition pressure, laser power, laser wavelength, number of ablation pulses and pulse repetition rate. X-ray diffraction analysis of the ZnO films deposited in various conditions was performed using a Bruker D2 Phaser (benchtop) (Billerica, Mass., USA). The roughness of films deposited in various conditions was determined using a SURFCOM 180 A (Tokyo Seimitsu, Tokyo, Japan) profilometer.

In order to establish the dependence of the SAW sensor properties on the morphology of the ZnO and Pd thin films, components of the sensitive bilayer, ZnO layers and Pd/ZnO bilayers were deposited onto SAW sensor substrates. The SAW sensors were of the delay line type and consisted of two port resonators with 50 electrode pairs and a periodicity of 11 µm. Interdigital transducers (IDTs) are obtained using standard photolithographic techniques on ST-X quartz substrates. IDTs have a periodicity of 45 µm and a 2500 µm wide acoustic aperture, and consist of a 10 nm chromium layer, which has good adherence to the quartz substrate, on top of which there is a 150 nm thick gold layer [27]. The thin film depositions onto the sensor substrates were made through a custom mask that delimited the sensing film to the area between the two electrode pairs. In the case of the sensors, the ZnO layers were deposited in 2 h at 10 kHz, while the Pd layers were thin, deposited in 2′23″ also at 10 kHz.

The sensors were characterized, as shown in Figure 1, using a CNT-91 Pendulum counter analyzer (Spectracom Corp, Rochester, NY, USA) with Time View 3 software (Pendulum Instruments, Banino, Poland). The sensor responses (frequency shifts) for various hydrogen concentrations between 0.2 and 2% were determined in a total gas flow rate maintained constant at 0.5 L/min in all cases. The various concentrations were obtained by combining a hydrogen gas mixture (2% H_2_/98% synthetic air) with pure synthetic air, using a system that included mass-flow meters and controllers. For more details on the experimental setup used for sensor characterization, see [28].

## 3. Results and Discussion

### 3.1. Film Morphology

#### 3.1.1. ZnO Films

The dependence of the morphology of ZnO films deposited onto Si substrates on various deposition parameters was determined based on SEM images.

The effect of deposition pressure on the morphology of the ZnO films is illustrated in Figure 2.

Of the oxygen pressures used, the ZnO film deposited in 700 mTorr O_2_ was the most porous, as the SEM cross-section images confirm (Figure 3). Although the images of the ZnO films deposited using 400 and 700 mTorr are qualitatively similar, the film deposited in 700 mTorr is clearly more porous. On the other hand, the films obtained in 100 and 400 mTorr are qualitatively different, as evidenced by Figure 2a,b. A comparison of the surfaces of the ZnO films deposited in 100 and 700 mTorr O_2_ was made using the Image J software [29]. The nanoparticles on the surface of the film have a mean diameter of approximately 21 nm (determined by measuring over 100 nanoparticles) in both cases. In the case of the film deposited in 700 mTorr, this is the dimension of the separated nanoparticles, not that of the agglomerations that form the porous nanostructures visible in Figure 2c, and in the cross-section in Figure 3. The difference between these two SEM images consists of the percentage of the area covered by the porous formations of nanoparticle agglomerations. In the case of the film deposited at high pressure, approximately 40% of the area of the surface in Figure 2c is covered by these formations, while at low pressures (Figure 2a), only approximately 6% of the surface consists of nanoparticle agglomerations. The larger porosity of the films deposited at higher pressures was confirmed by measurements of its surface roughness using a profilometer. The arithmetical mean deviation R_a_ and the root mean square deviation of the profile R_q_ of the ZnO film deposited using 1.06 µm radiation in 700 mTorr O_2_ were found to be 21.00 and 26.67 nm, respectively. In the case of a film deposited using the same wavelength but in 100 mTorr, R_a_ = 15.67 nm and R_q_ = 20.33 nm. As can be observed in Figure 3, the large porosity of the film implies that the thickness of the film is non-uniform. The average thickness of the film in Figure 3, which was deposited in the same conditions as films used in SAW sensors, is approximately 850 nm. It should be noted that we did not attempt to use higher deposition pressures since they lead to films that are unstable on the substrate surface, their adherence being poor.

The fact that higher deposition pressures lead to increased film porosity can be explained through the hydrodynamic effects of the interaction between the laser-ablated species and the ambient gas at relatively high pressures. These effects lead to a pronounced slowing of the species in the ablation plasma. In these conditions, the ablated species are confined in the target–substrate region by the interaction with the gas molecules, the plasma density increases, and increased collisions between the ablated species lead to nucleation in the gas phase, with the formation of relatively slow nanoparticles in the target–substrate region [30,31,32,33].

This pressure dependence of the film morphology should be reflected in the properties of the sensors, since it is known that more porous films lead to better sensor responses [16]. We therefore measured the sensing properties of SAW sensors that incorporate ZnO layers deposited with 100 and 700 mTorr, in order to ascertain the role of the porosity of the ZnO component of the bilayer on the sensor response.

The morphology of the films changes as the thickness increases, as has been observed in other cases when depositions are made at high pressures [25,27]. The dependence of morphology on thickness is illustrated by the SEM images of the surfaces of ZnO films deposited in 2 h and 3 h (Figure 4a,b, respectively) using a laser wavelength of 1 µm, a repetition rate of 10 kHz and a deposition pressure of 700 mTorr O_2_. As has been previously observed, the morphology of the film varies with the number of pulses [34]. The morphology of the films is not uniform over their entire thickness, with the films becoming more porous as the number of pulses increases. Films were not deposited with a larger number of pulses, since the acoustic wave attenuation is too high when the sensitive film is too thick.

The ZnO films also become more porous with increasing laser power for powers between 0.1 and 0.3 W. The laser pulse repetition rate being the same in all cases (10 kHz in the case discussed here), the energy per pulse increases with increasing power, being of the order of tens of µJ. The differences in morphology can probably also be explained by the variation in thickness with power.

Regarding the dependence of the film morphology on wavelength, the difference between ZnO films deposited at 532 nm and 1 µm is not considerable (Figure 5a,b). However, at a large scale, the films deposited at 532 nm seem to have a larger density of porous formations than the ones deposited at 1.06 µm (Figure 5c,d). This will be reflected in the properties of the sensors into which they are incorporated, as will be discussed later.

ZnO films were deposited at two different laser repetition rates: 10 kHz, as used in the rest of the depositions with the ps laser, and 50 kHz. By varying the average power of the laser correspondingly (0.2 W for 10 kHz and 1 W for 50 kHz), the energy per pulse can be maintained in both cases, namely 20 µJ/pulse. In this domain, the repetition rate had no effect on the morphology of the film. It should be noted that 50 kHz was the limit of the domain attainable, since, although larger differences in repetition rates were possible, the average power necessary to ensure the same energy/pulse could not be reached.

As is visible in Figure 6, depositions of ZnO films using a laser with ps pulse durations led to considerably more porous ZnO layers than those with a ns laser. However, comparison between the effects of laser pulse duration is difficult due to the large differences that exist in other parameters: energy/pulse and pulse repetition rates. The energy per pulse is approximately 30 µJ per pulse for the ps laser and 75 mJ/pulse for the ns laser, while the pulse repetition rate is 10 kHz and 10 Hz, respectively, which means a difference of three orders of magnitude. The differences in average and peak powers are, however, not particularly large: 0.3 W average power and 7.5 × 10^6^ W peak power for ps, and 0.7 W average power and 13 × 10^6^ W peak power for the ns laser. The difference is therefore, in this case, only a factor of around two. Regardless of the difficulty of comparing the effects of the two lasers, it is possible to state that from a practical standpoint, with the ps laser, it is easier to obtain porous ZnO films than with the ns laser. At the 532 nm emission wavelength, the ns laser does not produce porous structures, even at the relatively high deposition pressure of 700 mTorr, while the ps laser produces porous structures even at 400 mTorr.

XRD analysis of the ZnO films shows that no Zn lines are present, indicating correct ZnO stoichiometry. A crystallite size of 48 nm was determined by applying the Scherrer formula to the (101), (002) and (100) peaks of ZnO (according to diffraction pattern 2107059) present in the diffraction data obtained for a film deposited using IR radiation, in 700 mTorr O_2_.

#### 3.1.2. Pd Films

In the case of Pd thin films, the dependence on pressure, laser power and number of pulses was investigated, and a comparison of film morphology obtained with the two different laser wavelengths was made.

As in the case of ZnO, the porosity of the Pd films increases with deposition pressure. Figure 7 presents the surfaces of Pd thin films deposited in 30′ at 10 kHz. At this number of pulses, the surface of the layer deposited in 700 mTorr Ar appears slightly more porous than the one deposited in 100 mTorr. A cross-section SEM image reveals the large porosity of the Pd thin film deposited in 700 mTorr. As in the case of ZnO, the porosity of the film leads to large variations in the local film thickness across its surface. We estimate that the thickness of the thin Pd film used in the sensors is approximately 25 nm.

At the smaller number of pulses used to deposit the thin Pd layer in the SAW sensors (2′23″ at 10 kHz), the difference between layers deposited at 100 and 700 mTorr is not visible in the SEM images. As we will discuss in the section describing the sensor properties, however, some difference between the sensors comprising thin Pd layers deposited at 100 and 700 mTorr does exist in the frequency shift produced in the presence of H_2_. As we have previously mentioned, the Pd component of the sensitive film bilayer of the SAW sensor is thinner than the ZnO layer, because its main role is to dissociate the hydrogen molecules, thereby increasing the mobility (diffusion) of the detected species into the sensing layer, which leads to an increase in the sensor’s sensitivity [26]. Pd is therefore deposited with a smaller number of pulses. Figure 8 presents SEM images of the surfaces of Pd thin films deposited with different pulse numbers.

Regarding the dependence on laser power, visible differences only appear at high deposition pressure (700 mTorr). As seen in Figure 9, the porosity increases with laser power. As is the case with ZnO, this is due to the increase in film thickness with laser power.

#### 3.1.3. Pd/ZnO Bilayer

SEM images of the bilayer surface (the surface of Pd deposited on top of the ZnO) were obtained for the same deposition conditions as those used for the SEM sensors. In the case of ZnO deposited in 700 mTorr O_2_, there is no clear difference between the bilayers with Pd deposited in 100 mTorr Ar, those deposited in 700 mTorr Ar or the surface of the ZnO layer only, as is visible, for example, in Figure 10. Similar results were obtained for all the cases investigated, indicating that the morphology of the relatively thin Pd layer does not affect the morphology of the bilayer surface. This is confirmed by the EDAX mapping results of the bilayer surface. The distribution of the Pd on the surface of the bilayer is the same for Pd layers deposited in 100 and 700 mTorr Ar, as well as for both the 532 nm laser wavelength and the 1.06 µm wavelength. EDAX also confirms the fact that the ZnO is deposited more uniformly across the surface of the sensitive thin film when IR laser radiation is used, and it presents areas of larger Zn and O concentrations in the case of the 532 nm wavelength. These areas of larger concentration correspond to the porous density formations discussed in relation to Figure 5d. The morphology of the bilayer is therefore practically determined by that of the underlying ZnO layer. EDAX maps can be found in the Appendix A.

### 3.2. Sensor Properties

The response to hydrogen of SAW sensors having sensitive layers with Pd/ZnO bilayers deposited in different conditions, as well as single ZnO layers, was determined. Table 1 gives the deposition parameters used to obtain the sensitive layers for the sensors that were tested. All of the sensitive layers were deposited using the laser with ps pulse durations (which leads to more porous films) using 10 kHz and a power of 0.2 W. The ZnO layers were deposited in 2 h, and the Pd layers in 2′23″.

The dependence of the sensor responses on the hydrogen concentration between 0.2 and 2% is demonstrated in Figure 11, for deposition wavelengths of 532 nm and 1.06 µm, respectively. The sensitivity and limit of detection (LOD) for all sensors realized are also listed in Table 2. The noise level was estimated at ~5 Hz for (S1, S5) and ~10 Hz for (S2, S3, S4, S6). The results indicate not only the importance of the Pd/ZnO bilayer in comparison to a single layer of ZnO, but also the importance of the morphology of the films that make up the bilayer, especially that of the ZnO film.

The sensors having a sensitive layer of ZnO only and those with Pd/ZnO bilayers, with the ZnO deposited in 100 mTorr oxygen, did not respond to low hydrogen concentrations. The difference in the response of sensors containing ZnO deposited at higher oxygen pressure, which are more porous, from those deposited in 100 mTorr oxygen is clearly visible, the porous films leading to a larger response, higher sensitivity and lower LOD. The relative difference between the response of sensors having ZnO layers deposited in 100 mTorr and 700 mTorr O_2_ is 66–70% for all hydrogen concentrations investigated, regardless of the overlying Pd layer properties or the laser wavelength used. This indicates the importance of the porosity of the ZnO layer in determining the SAW sensor response. The influence of the Pd layer morphology, on the other hand, is less evident, with the relative difference in the frequency shifts for sensors with Pd layers deposited in 100 or 700 mTorr Ar on top of ZnO layers deposited in identical conditions being only in the range of 12–20%. This is to be expected since the morphology of the thin Pd layers deposited at the higher and lower pressures used in the sensors is not qualitatively different, as the SEM results presented have shown. However, it is worth mentioning that it is always the Pd layers deposited at higher pressure that produce better results. The presence of a Pd layer on top of ZnO leads to a significant improvement in the sensor response in comparison to a ZnO layer only, as we have shown in previous research, as well [25]. The results given in the present paper indicate that the morphology of the ZnO component of the bilayer has an influence on the sensor properties of the same order as the influence of the presence of the Pd layer.

Regarding the dependence of sensor properties on the wavelength of the radiation used, the 532 nm wavelength leads to somewhat better results than the infrared wavelength, as can be seen by comparing Figure 10a,b (note the difference in vertical scale). This can be explained by the morphological differences in Figure 5, namely that the films deposited at 532 nm seem to have a larger density of porous formations than the ones deposited at 1.06 µm.

## 4. Conclusions

Previous research has indicated that a Pd/ZnO bilayer leads to improved SAW sensor properties compared to single Pd or ZnO layers [25]. The present paper describes results regarding the contribution of the morphology of the ZnO and Pd thin film components of the bilayer to the sensor response. These results were obtained using an important advantage of PLD, namely the relative ease with which the morphology of the layers can be controlled through variation of the laser deposition parameters. 

For both materials studied, as has been previously observed for other materials [16], higher deposition pressures led to more porous films. Larger porosities are also obtained in the case of thicker films and in those obtained with a larger energy per pulse. Regarding the dependence on the laser wavelength, ZnO films deposited using 532 nm have a larger density of porous formations than those deposited with IR radiation. Although comparison between lasers with ns and ps pulse duration are difficult due to differences in other parameters, ZnO depositions using ps pulse durations were, in the cases studied by us, considerably more porous than those obtained with ns pulses at the same deposition pressure.

The differences in the porosity of the films affect the responses of the SAW sensors into which they are incorporated, with the deposition conditions producing more porous films leading to a larger sensor response. The morphology of the ZnO component of the bilayer is decisive, with films deposited in 700 mTorr O_2_ using 532 nm having the largest responses (largest frequency shifts for a given hydrogen concentration). The effect of the Pd film morphology, although present, is smaller than that of ZnO, probably due to the smaller film thickness. However, the results obtained with Pd films deposited at higher pressures are systematically slightly better than those obtained at lower Ar pressures.

The main result of the research presented here is that the morphology of the ZnO component of the sensitive layer incorporated into the SAW sensor has an influence on the sensor properties of the same order of magnitude as the use of a bilayer instead of a simple Pd or ZnO layer. This could have implications in other bilayer material combinations used in such sensors and for other types of analytes.

## Figures and Tables

**Figure 1 nanomaterials-11-02598-f001:**
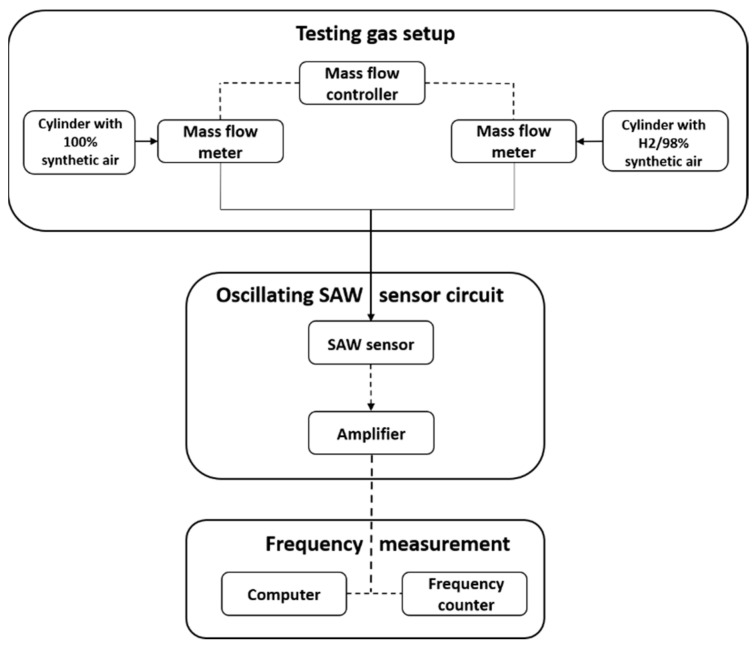
Experimental setup for SAW sensors’ frequency shift measurements for hydrogen detection.

**Figure 2 nanomaterials-11-02598-f002:**
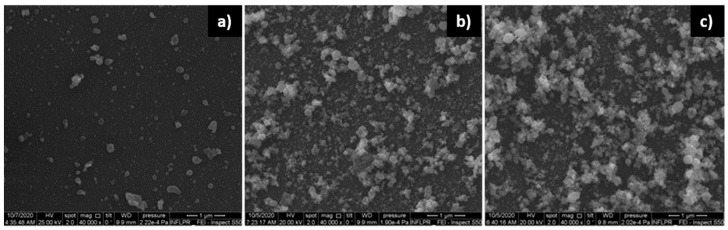
SEM images of ZnO films deposited using a ps laser, a wavelength of 532 nm, a power of 0.2 W and a pulse repetition rate of 10 kHz. The O_2_ pressure used was (**a**) 100 mTorr; (**b**) 400 mTorr; (**c**) 700 mTorr.

**Figure 3 nanomaterials-11-02598-f003:**
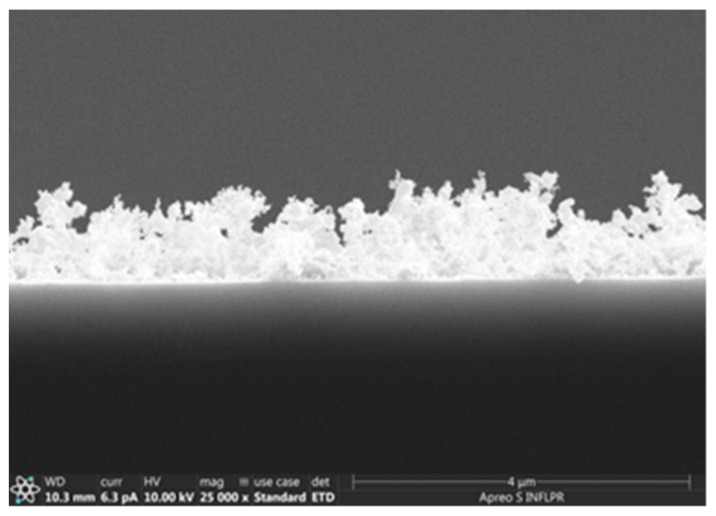
Cross-section SEM image of a ZnO film deposited in 2 h at a repetition rate of 10 kHz, a wavelength of 532 nm, using a laser power of 0.2 W, in 700 mTorr O_2_.

**Figure 4 nanomaterials-11-02598-f004:**
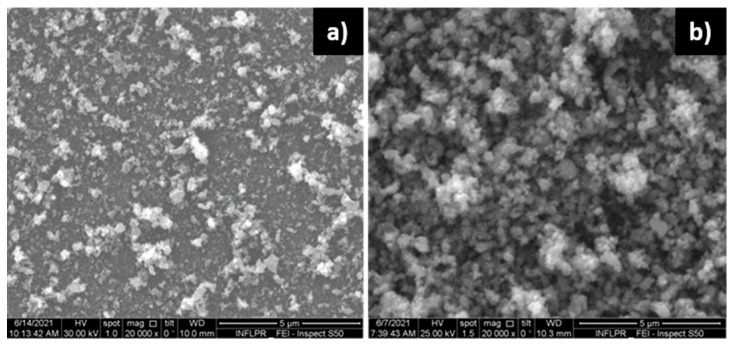
SEM images of ZnO films deposited using a laser wavelength of 1 µm, at a power of 0.2 W and a repetition rate of 10 kHz, in 700 mTorr O_2_, using (**a**) 2 h deposition time; (**b**) 3 h deposition time.

**Figure 5 nanomaterials-11-02598-f005:**
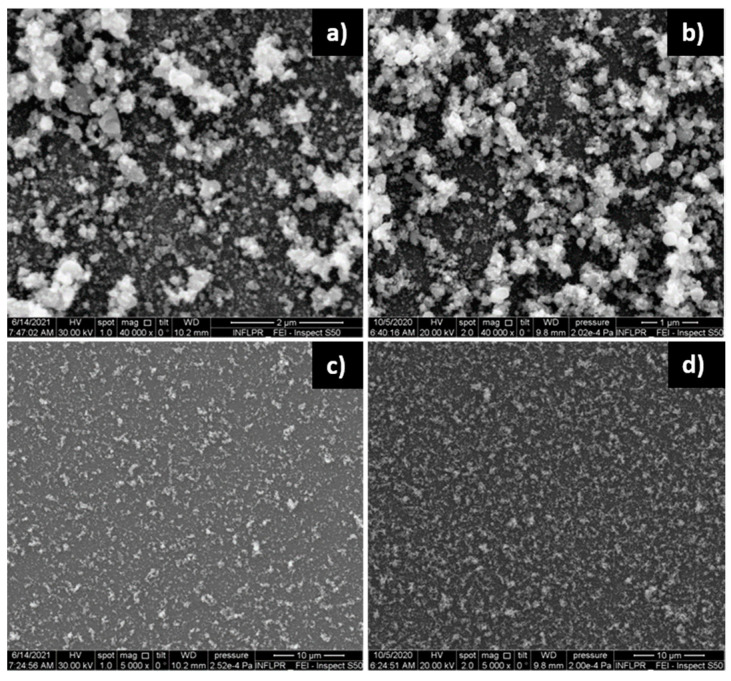
SEM images, at two magnifications, of ZnO films deposited in 700 mTorr O_2_ with 0.2 W using 1.06 µm (**a**,**c**) and 532 nm (**b**,**d**) laser emission wavelength.

**Figure 6 nanomaterials-11-02598-f006:**
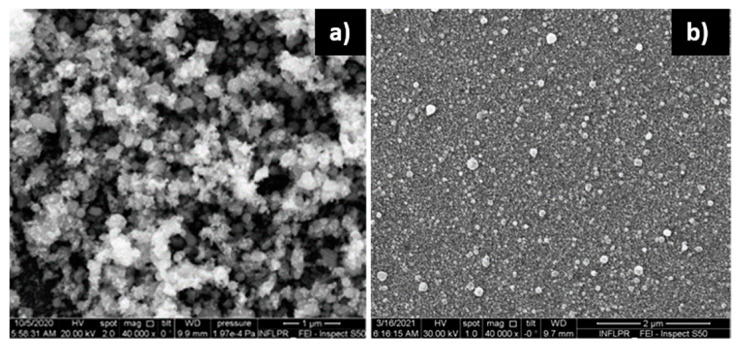
SEM images of ZnO films deposited in 700 mTorr O_2_ and a wavelength of 532 nm using (**a**) a ps laser with 0.3 W operating at 10 kHz repetition rate; (**b**) a ns laser with 75 mJ/pulse operating at 10 Hz repetition rate.

**Figure 7 nanomaterials-11-02598-f007:**
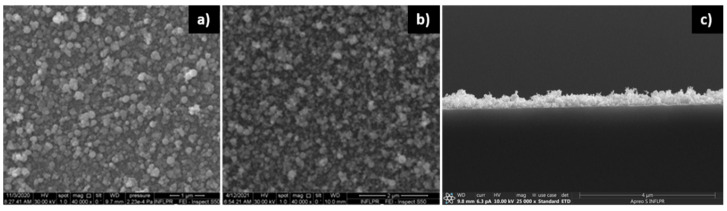
Pd thin films deposited at 10 kHz in 30′ using the 532 nm laser radiation of a ps laser, and an average power of 0.2 W. (**a**) in 100 mTorr Ar; (**b**) in 700 mTorr Ar; (**c**) cross-section of film deposited in 700 mTorr Ar.

**Figure 8 nanomaterials-11-02598-f008:**
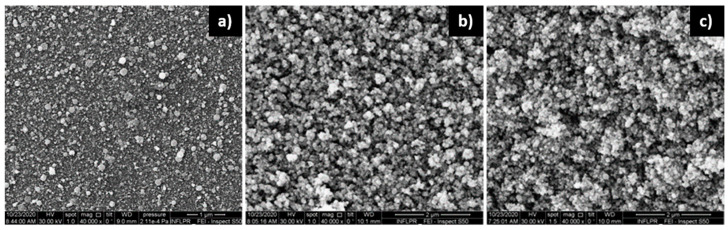
Pd films deposited at 10 kHz with a ps laser with 0.3 W and 532 nm, in (**a**) 2′23″; (**b**) 30′; (**c**) 2 h. The deposition pressure was 400 mTorr Ar.

**Figure 9 nanomaterials-11-02598-f009:**
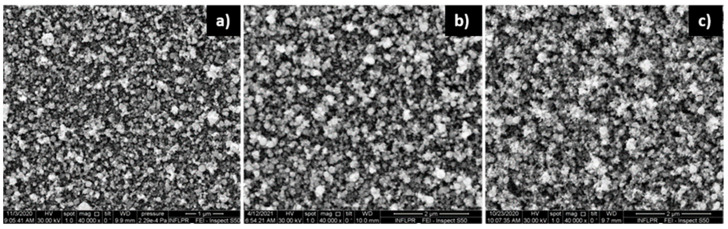
SEM images of the Pd thin film surface. Films were deposited in 700 mTorr Ar with a ps laser, using 532 nm radiation, at 10 kHz, in 30′, using an average power of (**a**) 0.1; (**b**) 0.2 W; (**c**) 0.3 W.

**Figure 10 nanomaterials-11-02598-f010:**
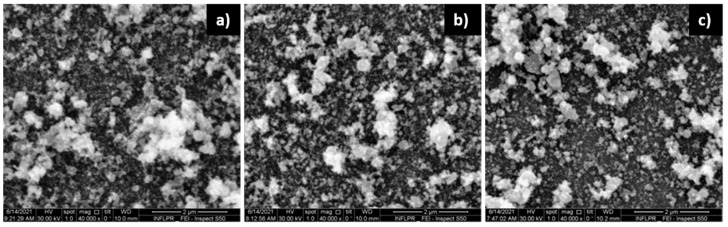
SEM images of (**a**) Pd/ZnO bilayer deposited in 2′23″/2 h with 10 kHz, using 0.2 W laser power at a wavelength of 1.06 µm. The Pd layer is deposited in 100 mTorr Ar, and the ZnO layer in 700 mTorr O_2_; (**b**) same as (**a**), but the Pd layer deposited in 700 mTorr Ar; (**c**) ZnO layer only deposited in the same conditions as in (**a**,**b**).

**Figure 11 nanomaterials-11-02598-f011:**
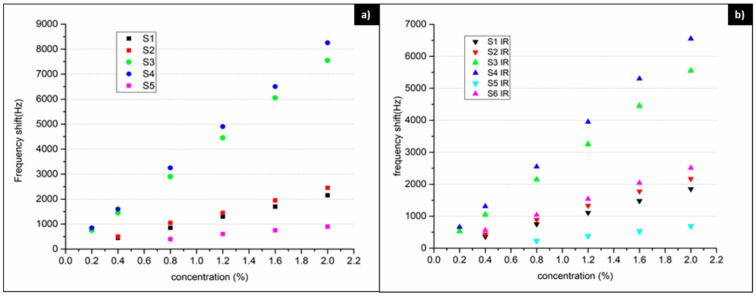
Sensor response to various H_2_ concentrations. (**a**) Sensitive films deposited with 532 nm laser radiation. (**b**) Sensitive films deposited with 1.06 µm laser radiation.

**Table 1 nanomaterials-11-02598-t001:** Conditions for deposition of the components of the sensitive films for the sensors that were measured. Sensors S5, S5 IR and S6 IR have a sensitive layer consisting in a single ZnO layer.

Sensor	Deposition Wavelength (nm)	O_2_ Pressure—ZnO Deposition (mTorr)	Ar Pressure—Pd Deposition (mTorr)
S1	532	100	100
S2	532	100	700
S3	532	700	100
S4	532	700	700
S5	532	100	-
S1 IR	1064	100	100
S2 IR	1064	100	700
S3 IR	1064	700	100
S4 IR	1064	700	700
S5 IR	1064	100	-
S6 IR	1064	700	-

**Table 2 nanomaterials-11-02598-t002:** Sensitivity and limit of detection (LOD) of the sensors. Legend: Δf—frequency change; c—hydrogen concentration; *n*—noise level.

	532 nm			1064 nm	
Sensor	Sensitivity (Δf/c) (Hz/ppm)	LOD (3×*n*)/ (Δf/c) (ppm)	Sensor	Sensitivity (Δf/c) (Hz/ppm)	LOD (3×*n*)/ (Δf/c) (ppm)
S1	0.108	138.67	S1 IR	0.0923	162.6
S2	0.124	241.37	S2 IR	0.1099	273.06
S3	0.371	80.85	S3 IR	0.2705	110.93
S4	0.41	73.22	S4 IR	0.3274	91.64
S5	0.048	312.7	S5 IR	0.032	468.6
			S6 IR	0.1303	230.3

## Data Availability

Not applicable.

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
