# Peer review of "Effect of Pd/ZnO Morphology on Surface Acoustic Wave Sensor Response"

_nanomaterials, 2021, doi:10.3390/nano11102598_

Round 1

Reviewer 1 Report

A hydrogen sensor on surface acoustic waves with a Pd/ZnO bilayer film as a sensitive layer was investigated in this work. It was shown that the sensitivity of the sensor depends on the morphology of the Pd/ZnO bilayer film. The morphology of the ZnO film is a determining factor. The sensitivity of the sensor increases with increasing porosity of the sensitive layers. The pulsed laser deposition method was used to form the films, which allows the formation of films with different morphologies. The morphology of the films is determined by the pulse repetition frequency, the wavelength of laser radiation, the distance from the target to the substrate, and the temperature of the substrate.

In general, the article is of a materials science nature.

The article is well written.

It should be noted that the content of the article is more suitable for the journal Sensors.

A number of critical remarks should be made about the content of the article:

(1) It is necessary to present in a separate figure the circuit of the SAW sensor and the circuit for measuring the amplitude-frequency characteristics of the sensor.

(2) It is reasonable to present the amplitude-frequency characteristics of SAW sensors for bilayer films obtained under different conditions. On the one hand, the porosity of the films increases the sensitivity of the sensor, but on the other hand, it can negatively affect the SAW propagation process, since porosity can lead to distortion of the SAW wavefront and the corresponding change in the amplitude-frequency characteristics.

(3) For the most sensitive sensor (S4) it is necessary to present the amplitude-frequency characteristics of the sensor for different hydrogen concentrations, which will be a clear illustration to Figure 10.

(4) It is necessary to determine the limit of the sensor performance, because hydrogen binding in ZnO and Pd has a limit. And then how to restore the sensitivity of the sensor?

Author Response

Dear Reviewer 1,

We thank you for the time and interest you have given to this work and for the desire to clarify certain important notions. We attached you the responses.

Reviewer 2 Report

The article is devoted to the synthesis of ZnO/Pd films for SAW gas sensors. The main emphasis is placed on the analysis of the classical dependence between the synthesis conditions, the morphology of the films and their sensor properties in hydrogen detection.

The weak side of the article is a qualitative description of the parameters of the microstructure at the level of "more porous - less porous". In order for the article to become suitable for publication, it is necessary to provide quantitative estimates of the parameters of the microstructure.

  1. For example, even from the SEM data available in the article, it is possible to estimate the sizes of particles, which forme the films.
  2. From the X-ray diffraction data (it is necessary to effectuate such a study), it is possible to estimate the average particle size of the crystal phases, at least for ZnO. In addition, it is possible to determine the presence/absence of film textures from XRD.
  3. It is necessary to characterize the structure of the films by the AFM method. This will allow the authors to determine the surface roughness of the films, which is indirectly related to their porosity.
  4. Finally, it is necessary to conduct a study of the films containing Pd layer by the XPS method. This will allow the authors to determine whether there is an influence of the conditions of Pd deposition on its chemical state (the degree of palladium oxidation). This characteristic is not related to the morphology of the films, but it affects the activity of palladium in hydrogen spillover. The use of ion etching in the XPS experiment will allow to determine the distribution of palladium over the thickness of the ZnO layer.

Author Response

Dear Reviewer,

We thank you for the time and interest you have given to this work. We attached you the responses.  

Round 2

Reviewer 1 Report

The article may be published as amended. The changes made are correct.

Author Response

Thank you.

Reviewer 2 Report

The authors need to check the particle sizes obtained by the SEM and XRD methods. According to SEM data, the particle size is 21 nm, according to XRD data, the crystallite size is 48 nm. This, at least, is strange. Since SEM is a direct method for determining the size, we can assume an error in estimating the cerystallites size using the Scherrer formula. It is necessary to make an estimate of the size from the broadening of the reflections separately for the diffraction maxima (101), (002) and (100).

Author Response

The difference in the dimensions is due to the difference in what is measured. As we specified in the text referring to the SEM images in figure 2:

“The nanoparticles on the surface of the film have a mean diameter of about 21 nm (determined by measuring over 100 nanoparticles) in both cases. In the case of the film deposited in 700 mTorr, this is the dimension of the separated nanoparticles, not that of the agglomerations which form the porous nanostructures visible in Figure 2c, and in cross section in Figure 3.”

We could only measure the dimensions of the nanoparticles which are separated in the SEM images, not the dimensions of those in the agglomerations, where nanoparticles cannot be clearly separated. The nanoparticles which were measured in SEM images form at the substrate surface in the initial part of the film deposition. After a larger number of pulses, the morphology of the deposited film changes from separated nanoparticles to the porous agglomerations visible in figures 2 and 3. As we have specified in the text:

“The morphology of the films changes as the thickness increases, as has been observed in other cases when depositions are made at high pressures [25, 27]. The dependence of morphology on thickness is illustrated by SEM images of the surface of ZnO films deposited in 2 h and 3 h (Figures 4 a) and b), respectively) using a laser wavelength of 1 µm, a repetition rate of 10 kHz, and a deposition pressure of 700 mTorr O2. As has been previously observed, the morphology of film varies with number of pulses [34]. The morphology of the films is not uniform over their entire thickness, with the films becoming more porous as the number of pulses increases.”

The porous agglomerations, which cover a larger part of the film surface than the separated nanoparticles, are the ones detected by XRD, and these determine the dimensions of the crystallites determined from XRD data. The nanoparticles which can be measured in SEM images are separate nanoparticles formed in the initial stages of the deposition, which are smaller than those in the agglomerations.

We mention that the widths of the diffraction maxima at (101), (002) and (100) are 48, 56 and 42 nm respectively.